# Integrated Precision High-Frequency Signal Conditioner for Variable Impedance Sensors

**DOI:** 10.3390/s24206501

**Published:** 2024-10-10

**Authors:** Miodrag Brkić, Jelena Radić, Kalman Babković, Mirjana Damnjanović

**Affiliations:** Faculty of Technical Sciences, University of Novi Sad, Trg Dositeja Obradovića 6, 21000 Novi Sad, Serbia; brkic.miodrag@uns.ac.rs (M.B.); bkalman@uns.ac.rs (K.B.); mirad@uns.ac.rs (M.D.)

**Keywords:** CMOS technology, sensors, signal conditioning

## Abstract

In this paper, a signal conditioner intended for use in variable impedance sensors is presented. First, an inductive linear displacement sensor design is described, and the signal conditioner discrete realization is presented. Second, based on this system’s requirements, the integrated conditioner is proposed. The conditioner comprises an amplifier, a tunable band-pass filter, and a precision high-frequency AC-DC converter. It is designed in a low-cost AMS 0.35 µm CMOS process. The presented conditioner measures the sensor’s impedance magnitude by using a simplified variation of the sensor voltage and current vector measurement. It can be used for the real-time measurement of fast sensors, having small output impedance. The post-layout simulation results show that the integrated conditioner has an inductance measurement range from 10 nH to 550 nH with a nonlinearity of 1.2%. The operating frequency in this case was 8 MHz, but the circuit can be easily adjusted to different operating frequencies (due to the tunable filter). The designed IC area is 500 × 330 μm^2^, and the total power consumption is 93.8 mW.

## 1. Introduction

Variable impedance sensors are widely used in various electronics systems and applications, such as industrial applications (e.g., monitoring and process control systems, manufacturing automation), robotics, healthcare, household appliances, commercial devices, etc. The rapid development of new technologies and the discovery of new and improved sensor materials and methods enable sensor size and cost decrease and performance improvement (e.g., better accuracy and sensitivity). Cutting-edge sensor generations with reduced output impedance (usually several orders of magnitude) require new, high-performance sensor signal conditioners.

Simple signal conditioners based on measurement (mainly the Wheatstone) bridges are very precise and usually used in national laboratories for measurement on standards (etalons) [1]. They are rarely used outside laboratories and in sensors for continuous measurements since they need manual adjusting in every measurement, leading to a very poor dynamic response.

Auto-balancing bridges utilizing analog electronic circuits can be found in impedance analyzers and LCR meters, which use different measurement methods; nevertheless, they are mostly based on sensor voltage and current vector measurement [1]. They are characterized by high accuracy and wide measurement range; yet, these features are obtained using a complex design, so these systems are regarded as having a high cost, with large dimensions and high current consumption. Therefore, they are not suitable for use in measurement systems intended for mass production.

In conditioners based on digital automatic measuring bridges, all signal processing is performed in the digital domain, eliminating the need for sophisticated analog electronic circuits. DSP processors are used for the implementation of the virtual balancing of measurement bridges. With these algorithms, one-half of the bridge is virtually “generated” in the digital domain, which provides a very accurate measurement (accuracy up to 0.02%) [2]. Various algorithms are used to generate the virtual part of the bridge, such as “least mean square” [3,4], “sine-fitting” [2], and “ellipse-fitting” [5,6] algorithms. This method enables the development of high-precision electronic systems, though usually at low frequencies. An increase in the system excitation frequency increases the system hardware complexity due to the need for faster A/D converters and more powerful processors, which significantly increases their production costs.

Conditioners that rely on LC oscillators [7,8,9,10] are simple solutions for measuring inductance or capacitance because there is a strong dependence between the sensor impedance and the output signal frequency that can be measured easily and precisely. The main drawback of this approach is accuracy (highly dependent on the quality factor of the resonant circuit) [11], as it is difficult to reduce the frequency dependence on temperature due to problems related to the oscillation stability. In addition, sensors must have high output impedance and high-quality factors [12]. The method is often applied in wireless sensors inductively coupled with electronic measuring systems [13,14,15].

Conditioners that use lock-in amplifiers (or synchronous detectors) can also be found in the literature. Lock-in amplifiers are generally used to measure small signal amplitudes in high-noise environments. They are based on a homodyne detector followed by a low-pass filter. Conditioners with analog lock-in amplifiers can provide high accuracy [16], though at the expense of circuit complexity, as they require high-quality sine mixers whose complexity significantly increases at high frequencies (thus often used only at lower frequencies) [17,18]. Accurate and simple low-cost analog mixers can be designed for low operating frequency (below several hundred KHz) [19] and are usually used in laboratory measurement equipment. Simple conditioners based on square wave mixers (composed of analog switches) can operate at very high frequencies but achieve reduced measurement accuracy due to the significant mixer switching noise [20,21].

In electrical impedance tomography (EIT), many aspects of a person’s health status can be found by measuring real and imaginary components of the body impedance. In EIT impedance conditioners, many variations in synchronous detectors are used to eliminate various sources of noise. These conditioners usually measure impedance from around hundreds of Ω to hundreds of kΩ [22], which is the range of bioimpedance of human tissue. The measurement range is mostly in the range of kHz (where A/D converters are usually used instead of synchronous detectors to find impedance), though for cancer diagnosis, it can go up to several MHz [23].

The focus of this paper is the design of the signal conditioner for variable impedance sensors, although an inductive displacement sensor was realized and utilized in the experiment. The paper applies a simplified variant of the current–voltage measurement method. The method was first tested and proved using a discrete electronic measurement system, which showed satisfying measurement results but exhibited significant sensitivity (especially to temperature). To provide better system performance, the integrated signal conditioner was designed using a low-cost AMS CMOS process. In addition, a tuning circuit for adjusting the designed filter characteristics and minimizing the influence of the PVT variation is proposed. The tunable filter allows the integrated signal conditioner to be used at different frequencies.

The rest of the paper is organized as follows: Section 2 presents the proposed measurement method and the signal conditioner discrete realization. Section 3 describes in detail the integrated conditioner and its building blocks. The measurement results of the conditioner discrete realization and the post-layout simulation results of the proposed integrated conditioner are presented in Section 4, and Section 5 summarizes the paper’s main conclusions.

## 2. Conditioner Discrete Realization

The initial motivation for designing the conditioner proposed in this paper was to create a measurement system for a displacement sensor (displacements smaller than 1 mm) with an inductive output. The sensor was realized as a planar coil on a printed circuit board (PCB). It consists of two parts with a meander coil on each part, while one of the coils (Coil 2) is short-circuited, Figure 1b. The sensor inductance *L_S_* was measured between the terminals of Coil 1 (Figure 1a) [24]. When Coil 2 is shifted along the *x*-axis above Coil 1, their coupling is changed, and so is the sensor inductance *L_S_*, which is proportional to the displacement. Since the sensor element has a very small inductance value, two versions were implemented, the larger one with an inductance range from 300 nH to 400 nH, and the smaller one with an inductance range from 50 nH to 100 nH.

The proposed discrete conditioner is based on a measuring method, which is a simple variation in the voltage and current vector measurement on a sensor. Instead of two vector voltmeters used in impedance analyzers, one precision high-frequency AC/DC converter is used in the proposed conditioner, which measures voltage change on the sensor, excited with a high-frequency sine source. The voltage signal on the sensor is amplified and filtered at the operating frequency of the AC source. The block diagram of the proposed conditioner is shown in Figure 2. It is realized using discrete components and commercially available integrated circuits [24].

Since very small changes in inductance are measured, the AC source frequency should be very high in order to obtain a higher voltage value on a sensor. To avoid the skin effect that has a significant influence on the sensor element (planar PCB traces) at frequencies above 15 MHz, an AC voltage source at 8 MHz is chosen. A Direct Digital Synthesis (DDS) integrated circuit is used as an AC voltage source. While the DDS circuit generates very precise and stable sine signals, it also produces harmonics at its clock frequency (here 75 MHz), so a low-pass filter has to be used to eliminate the unwanted signal’s components.

The AC voltage source is connected to the voltage divider, consisting of resistor *R_r_* and equivalent sensor impedance *Z_S_* (shown in Figure 2). The corresponding transfer function of the circuit is:(1)$$A_{v}=\frac{v_{s}}{v_{i}}=\frac{Z_{s}}{R_{r}+Z_{s}}$$
where *v_s_* is the voltage measured on the sensor element, and *v_i_* is the AC source sine voltage.

From Equation (1), the impedance *Z_S_* can be determined as:(2)$$Z_{s}=\frac{R_{r}\cdot \left(V_{s\_eff}/V_{i\_eff}\right)}{1-\left(V_{s\_eff}/V_{i\_eff}\right)}$$
where *V_s_eff_* is the effective voltage on the sensor element, and *V_i_eff_* is the effective voltage of the voltage source.

A four-wire voltage measurement technique is used on the sensor element. In this measurement technique, the voltage needs to be measured by a differential amplifier. Since the sensor coils also behave as an antenna, a significant external noise can be added to the measured signal. To eliminate the interference, a band-pass filter (BPF) with a center frequency that is the same as the AC voltage source (8 MHz) is added. The filter is designed as a two-pole active filter, realized with operational amplifiers (op amps). An RMS detector, the AD8361 circuit manufactured by Analog Devices (Wilmington, MA, USA), is used as a high-frequency AC/DC converter. The output of this circuit is a DC voltage proportional to the RMS value of its input signal. Thus, the output DC signal is directly proportional to the sensor voltage signal, and the conditioner measures the sensor impedance magnitude.

The prototype sensor with the measurement setup is displayed in Figure 1b. The fixed Coil 1 of the sensor element is connected to the measuring system. The Coil 2 of the sensor element is connected to the platform of an optical microscope, which can be moved. Keitley 2410, manufactured by Keithley Instruments (Solon, OH, USA), a very precise instrument for generating and measuring DC voltages and currents, is used to measure the DC voltage at the output of the RMS circuit.

## 3. Integrated Conditioner

To improve system performance, especially to minimize the size and power requirements, the signal conditioner is designed in low-cost AMS 0.35 µm CMOS technology. This technology offers a higher voltage swing due to a higher supply voltage (3.3 V) and fewer design challenges (e.g., process variation) compared with advanced process nodes.

### 3.1. Operational Amplifier Design

All amplifiers used in the integrated conditioner are full differential (FD) operational amplifiers. FD op amps are more suitable since they have many advantages over single-ended output op amps: twice as large full-scale output, 3 dB higher signal-to-noise ratio, lower common-mode noise, and most importantly, reduced harmonic distortion since the signal second-order distortions are rejected by the FD op amp [25]. Moreover, the FD op amp can work at higher frequencies than its counterpart single-ended output op amp because it has smaller parasitic capacitances. The drawback of using the FD op amp is higher current consumption and the need for additional common-mode feedback (CMFB) circuitry to establish the output signals’ common-mode levels. The main objective of the op amp design is to achieve the highest possible gain (in an open-loop system) at the system’s working frequency while having a wide input and output voltage range and good stability. To provide simplicity and a small number of non-dominant poles in the transfer function, ensuring a very high GBW, telescopic FD op amp topology is chosen. A folded cascode amplifier is used to provide a higher voltage range (though with slightly worse other circuit characteristics).

The schematic of the folded amplifier is presented in Figure 3. The PMOS transistors (M1 and M2) are selected at the amplifier input to allow NMOS transistors to be used in the output section, thus ensuring a large *GBW*. To obtain high gain, cascode stages consisting of transistors M7–M10 are used as an active load. To provide the low output impedance and high current capabilities, buffer stages (the standard common-drain or source follower amplifier) are added to the output of the proposed cascode folded amplifier. Transistors M5 and M6 determine the bias current of the input differential pair (transistors M1 and M2), and their gates are connected to the CMFB circuit output.

The schematic of the designed common-mode feedback circuit is displayed in Figure 4.

There are several commonly used topologies in the CMFB circuit design, differing in the method of measuring the common output voltage. In this paper, a CMFB circuit with resistors (R_1_ and R_2_) at the input is applied since it provides the most linear transfer function with the highest input voltage swing. The CMFB circuit is a simple differential amplifier that compares common voltages on the folded amplifier output nodes (the gate of the transistor Mc1) with reference voltage *V_CM_* (the gate of the transistor Mc2). To achieve negative feedback, the output of the CMFB circuit controls the lower current mirror of the folded amplifier, and the output V_cmfb_ is connected to the folded amplifier via gates of transistors M5 and M6 (shown in Figure 3). The IN1 and IN2 inputs of the CMFB circuit are connected to the outputs of the folded amplifier.

### 3.2. Filter Design

The filter is designed in MOSFET-C topology, which enables the filter characteristics (usually the cut-off frequency) to be adjusted using an external reference. This is required for CMOS-integrated filters to compensate for changes due to process, voltage, and temperature (PVT) variations (especially the high tolerances of passive components) [26]. In this paper, the Tow–Thomas (TT) biquad filter is used. By replacing all pairs of resistors with adequate NMOS transistors in the standard RC filter realization, the MOSFET-C version of the TT-biquad is obtained (Figure 5). Due to the use of a fully differential op amp, this filter requires one less operational amplifier (just two FD amplifiers) than a single-ended output filter [27]. The filter can be used either as a band-pass or low-pass filter (LPF), depending on whether the output signal is taken from the V_BP_ or V_LP_ output. The sensitivity of the poles and other filter parameters to the variations in its passive component values is the smallest possible, less than one, and is proportional to the passive LCR filter sensitivity (minimal achievable sensitivity in practical filter design) [28].

The center frequency of the band-pass filter *ω*_0_ is calculated from (3), the quality factor (*Q*) from (4), and the gain *A* at the center frequency (*ω*_0_) from (5)
(3)$$\omega_{0}=\frac{1}{RC}$$
(4)$$Q=\frac{R_{4}}{R}$$
(5)$$A(\omega_{0})=\frac{a_{1}Q}{\omega_{0}}=\frac{R_{4}}{R_{1}}$$
where *R*, *R*_1_, and *R*_4_ are the drain-source resistance *r_DS_* of the MOSFETs MR, MR1, and MR4, in Figure 5, respectively.

### 3.3. Filter Tuning Circuit Design

Integrated continuous-time filters require an additional auto-tuning circuit to adjust their parameters. Because of process variations, the filter RC time constant can have an accuracy of around 30%, where *R* represents the drain-source resistance of MOS transistors in the TT-biquad.

By controlling the drain-source resistance *r_DS_* of MR MOSFET in the MOSFET-C filter, the resistance *R* in (3) is controlled, and the center frequency can be set to the desired value. Although the capacitor C, which also determines the center frequency, has large process variations, usually up to 10%, its temperature coefficient is very small (30 ppm/K), so controlling the reference resistance is enough to compensate for *C* variations and to set the desired center frequency. A discrete external resistor is used as a reference since it is easy to obtain an inexpensive SMD resistor with very low-temperature coefficient values (below 10 ppm/°C).

The schematic of the tuning circuit is presented in Figure 6. The resistor R_EXT_ is an ideal resistor representing an external reference. The NMOS transistor MT1 has the same dimensions as the NMOS transistor MR in Figure 5. When this circuit is turned on, the op amp OPtune output voltage *V_c_* rises. This leads to a change in the transistor MT1 drain-source resistance value *r_DS_* until the voltage at the inverting op amp OPtune input equals the non-inverting op amp OPtune input voltage value, i.e., the voltage *V_CM_*. In this case, MOSFET MT1 resistance *r_DS_* is the same as the resistance *R_EXT_*. To suppress sudden changes in the control signal *V_c_*, a capacitor C1 is connected as feedback to the op amp OPtune, so it works as an integrator. The control signal *V_c_* is connected to the gates of MOSFETs MR, MR4, and MR1 in Figure 5, and their *r_DS_* resistance will be the same as *R_EXT_* resistance for well-matched transistors MT1 and MR.

The op amp OPtune is designed as a standard two-stage operational amplifier, where the first stage has a PMOS differential input pair with NMOS current mirror active load. The second gain stage is a common source amplifier having an active load. With this architecture, a large DC gain and large output range are easily obtained, with small power consumption.

### 3.4. Precision High-Frequency AC-DC Converter Design

The RMS detector is used as a precision high-frequency AC/DC converter in the discrete prototype realization. This circuit is intended for operation with signals of various shapes and provides an accurate RMS value at its output, depending on the shape of the input signal [29]. The CMOS technology is not the most suitable choice for the realization of high-frequency RMS circuits due to the need for translinear components. Since sine signal is used for sensor excitation in this conditioner, signal amplitude measurement can be performed with a less complex topology. A precision high-frequency AC-DC converter can be realized with an absolute value block and an averaging block.

#### Absolute Value Block Design

An absolute value block provides the absolute value of a sinusoidal signal at its input. The main problem in this circuit design is providing a large dynamic range of the input signal and small nonlinearity of the transfer function while operating at high frequencies. This paper proposes a new absolute value circuit to overcome these problems. The circuit is composed of a transconductance (G_m_) voltage current converter (conveyor), followed by a CMOS AB current rectifier. Using a current rectifier instead of a voltage rectifier enables an absolute value circuit to operate at higher frequencies.

The G_m_ voltage current converter topology is shown in Figure 7. This circuit is essentially a G_m_ amplifier with fixed transconductance *G_m_*, whose design is based on previous circuits [30]. The output current *i*_o_ is proportional to the input differential voltage *V_IN_*, while the *G_m_* of this circuit is inversely proportional to the resistor R_gm_ value. To make linear dependence of the differential input voltage on the output current, the gate-source voltage (*V_gs_*) values of MOSFETs differential pair M1G and M2G should be constant. The negative feedback provides constant drain currents of the transistors M1G and M2G, ensuring that the *V_gs_* voltage values of these MOSFETs are not influenced by the changes in the **V_IN_** voltage value. The negative feedback circuit for the transistor M1G (M2G) consists of the transistors M3G and M5G (M4G and M6G). All the changes in the Rgm resistor current values are mirrored to the transistors M3G and M4G.

The remaining circuit MOSFET transistors are part of the cascode current mirrors, which enable the current changes in transistors M3G and M4G to be transferred to the circuit output so that output current *i*_o_ is equal to:(6)$$i_{o}=\frac{4\cdot {V_{in}}_{PK}}{Rgm}$$

The Gm converter output is connected to the CMOS AB current rectifier, displayed in Figure 8. When the input current *i_IN_* value is positive, the transistor M3U is turned on while the transistor M4U is turned off; thus, all the input current flows through the resistor Rog. When the input current *i_IN_* value is negative, the transistor M3U is turned off, and the transistor M4U is turned on; so, all the input current flows through the resistor R_od_. The transistors M3U and M4U are biased to work in the weak inversion mode; consequently, they are turned on for the input current small values, below μA [31]. The differential output signal of this circuit is equal to:(7)$$v_{oDIFF}=v_{og}-v_{od}=V_{dd}-i_{in}R_{o}$$
where *v*_o*DIFF*_ is the difference between the voltages *v*_o*g*_ and *v*_o*d*_, for R_o_ = R_od_ = R_og_.

The total gain (*K*) of the absolute value block is calculated by combining Equations (6) and (7) and is equal to:(8)$$K=4K_{Gm}\frac{Ro}{Rgm}$$
where the parameter *K_Gm_* is the Gm voltage current converter gain. Process and temperature variations in the resistors Ro and Rgm do not have any influence on the circuit total gain since changes in the resistors Ro and Rgm values cancel each other out, which can be concluded from Equation (8).

The averaging block is very simple and consists of a pair of resistors and capacitors, which make low-pass filters (Figure 8). Since the conditioner operating frequency value is supposed to be higher than 1 MHz, even a simple one-pole passive low-pass filter is suitable for filtering the output signal AC component so that the differential DC signal *v*_o*DIFFAVG*_ is obtained. The averaging block components R_izg_-C_g_ and R_izd_-C_d_ are assumed to be external discrete components since their dimensions would be very large (in the IC). This is a common approach since this block also defines the conditioner dynamic response, which can be changed by a user depending on an application or the conditioner’s specific use.

## 4. Results and Discussion

By analyzing the conditioner discrete realization and the integrated conditioner, it can be noticed that the discrete implementation somewhat differs from the integrated design. The amplifiers used in the realized discrete prototype are standard single-ended output low-noise operational amplifiers, while the full differential amplifier is utilized in the integrated design. In addition, the integrated filter includes the tuning and the temperature stabilization circuit. In the discrete conditioner realization, the RMS detector circuit is used, while in the integrated design, a much simpler circuit is designed for AC/DC conversion with improved conditioner characteristics. Therefore, these differences will be reflected in the results of the proposed conditioner realizations.

### 4.1. Conditioner Discrete Realization Measurements

By comparing the measurement results of the conditioner discrete realization to the results of the inductive sensor element measured by Impedance Analyzer HP4194A, the maximum measurement error is found to be 3% [24], as shown in Figure 9. The linear response zone of the tested sensor for displacement measurement is around 0.6 mm, as shown by vertical dotted lines in Figure 9. The measurements are made with a smaller variant of the displacement sensor having the inductance *L_S_* value in the range of 50 nH to 100 nH. This sensor impedance *Z_S_* also consists of the resistive part *R_S_*, which changes from 0.62 Ω to 0.67 Ω with the displacement in the linear measurement range. By taking the fixed mean value of *R_S_* equal to 0.65 Ω, the maximal error from the average value of the resistance in calculating *L_S_* from *Z_S_* is only 0.35%.

The RMS measurement range of the AD8361 circuit is determined by its conversion error. According to the technical documentation, the conversion error increases significantly when the RMS input signal value is below 20 mV [29]. The maximum voltage value at the RMS circuit output is around 4.5 V (for 5 V power supply voltage). Since the RMS circuit gain is fixed at 7.5, the gain of the input differential amplifier is 2, the filter block has unity gain, and their input voltage range is from 10 mV to 300 mV RMS. The output voltage range is a DC signal from 150 mV to 4.5 V.

For the excitation signal (from the driver block) peak value of 1 V and the resistor value of 100 Ω, the inductance measurement range is from 55 nH to 920 nH when measuring a sensor with dominant inductive impedance.

The power consumption of the conditioner discrete realization is 0.55 W (for 5 V power supply voltage).

Due to the significant sensitivity to temperature (change of 5.2% within the temperature range of 0 °C to 70 °C), this conditioner is suitable for use only in an environment with small temperature variations. High sensitivity to temperature variations is estimated as a consequence of a simple filter topology used in this prototype. Small changes in the filter parameters have a significant impact on the filter operating frequency and, thus, conditioner gain.

### 4.2. Post-Layout Simulation Results of the Proposed Integrated Conditioner

The presented integrated conditioner is designed using a low-cost 0.35 µm AMS CMOS process and simulated using the Spectre Simulator from Cadence Design Systems (San Jose, CA, USA). The technology has a supply voltage of 3.3 V, enabling a wide conditioner dynamic range. For the parasitic extraction of the layout design, the extraction tool Cadence^®^ Quantus™ QRC Extraction Solution, provided by Cadence Design Systems (San Jose, CA, USA), was used. The integrated conditioner layout is shown in Figure 10. All differential topologies were matched using the common-centroid technique. The integrated circuit occupies a die area of 500 × 330 μm^2^.

#### 4.2.1. Operation Amplifier Post-Layout Simulation Results

The post-layout simulation results for the designed operational amplifier in a closed-loop configuration are provided in Figure 11. The DC open-loop gain is 67.7 dB, and the gain-bandwidth product (*GBWP*) is 510 MHz. The designed operational amplifier in an open-loop configuration is used in the filter design. The differential amplifier DC closed-loop gain is set to 2 (6 dB) with a resistors’ value of 4 kΩ in feedback. The 3 dB bandwidth is around 254 MHz to provide the integrated conditioner operation at frequencies higher than 8 MHz. The total power consumption of this block is 21.54 mW. This is mainly the result of a trade-off between the circuit’s power consumption and bandwidth (operating frequencies) in the used technology. The DC current values in the FD op amp have to be in the order of mA to obtain a high open-loop gain at 8 MHz and higher frequencies (i.e., high *GBWP*). Likewise, a low-cost old/higher process node offers a higher voltage swing, though it increases power consumption due to higher power supply voltage.

#### 4.2.2. Tunable Filter Post-Layout Simulation Results

A TT-biquad filter is first designed with a center frequency of 8 MHz (to compare with the conditioner discrete realization results), where the ideal RC component values are calculated from (3). The filter quality factor value of 1 (*Q* = 1) is chosen as a compromise since the high-quality factor value increases the filter’s selectivity but deteriorates the other filter parameters. Higher *Q* factor values give rise to higher sensitivity of the filter gain at the band-pass center frequency to PVT variations. The filter’s sensitivity linearly increases with the quality factor value (4), and the final *ω_t_* influence is proportional to the filter *Q* factor value. The filter gain at the band-pass center frequency is set to 1 (*A*(*ω*_0_) = 1 or 0 dB) by using (5), and it also minimizes dependence on PVT variations. As previously mentioned, besides the band-pass configuration used in this paper, the proposed filter can also be used as a low-pass filter (3 dB bandwidth around 10.3 MHz), Figure 12a.

To provide the operation of the integrated conditioner at different frequencies, the proposed band-pass filter center frequency can be adjusted by changing the value of the external resistor in the tuning circuit. In the case shown in Figure 12b, the filter center frequency is in the range of 5.43 MHz to 16.03 MHz for the resistor R_EXT_ value changing from 2 kΩ to 10 kΩ, though the frequency range is not limited to this one.

In addition to controlling the filter center frequency and thereby compensating for PVT variations, the proposed tuning circuit also decreases the dependence of the filter center frequency and the filter gain at the *ω*_0_ on the temperature changes. This represents a significant advantage in comparison to the conditioner’s discrete realization. When the filter is controlled by the tuning circuit, a variation in its center frequency is around 4.3%, from 7.77 MHz to 8.12 MHz, for a temperature range of 0 °C to 85 °C, while the filter gain at *ω*_0_ changes by 0.5% (Figure 12c). Variations without the tuning circuit are much larger, with a filter center frequency change of around 32.5% and a filter gain at the *ω*_0_ change of around 13%. The filter power consumption is 42.88 mW since it contains two operational amplifiers having a very wide bandwidth. Although Gm-C filters are a very popular choice in CMOS IC design because of their low consumption and high working frequency, they are not used in this circuit due to their high nonlinearity.

#### 4.2.3. Precision High-Frequency AC-DC Converter Post-Layout Simulation Results

The post-layout simulation result of the CMOS AC-DC converter output signal is shown in Figure 13. The positive half of a sine wave is displayed when the current flows through the resistor Rod (the voltage *V_od_*); within the negative half of a sine wave, the current flows through the resistor Rog (the voltage *V_og_*). The transfer function of the proposed AC-DC converter is shown in Figure 14. The maximum nonlinearity of the transfer function is 0.4% for the input voltage peak amplitude in the range of 20 mV to 1.1 V. The AC-DC converter’s maximum operating frequency is 117 MHz. The power consumption is 29.38 mW, as the DC currents in this circuit have to be in the order of mA to obtain a significant bandwidth (and thus to provide the operation of the integrated conditioner at higher frequencies).

#### 4.2.4. Integrated Conditioner Post-Layout Simulation Results

By analyzing the post-layout simulation results of the whole integrated conditioner (for the input voltage range determined during the analysis of the precision AC-DC converter), a higher nonlinearity is obtained, about 1.2%, compared with the nonlinearity of the AC-DC converter itself, which is 0.4%. Therefore, the nonlinearity of the proposed conditioner building blocks was analyzed, and the results indicate that the amplifier block has no significant nonlinearity, while the nonlinearity of the filter block increases sharply for high-amplitude signals. Since the total gain of the differential amplifier and filter is around 2, the input voltage range of the integrated conditioner is narrowed to values from 10 mV to 550 mV (in comparison to the AC-DC conditioner alone). The post-layout simulation result of the integrated conditioner output voltage dependence on the inductance value at its input is displayed in Figure 15. The circuit can measure the inductance value from 10 nH to 550 nH (covering the measurement ranges of both tested displacement sensors), while the nonlinearity is increased at higher values. The impedance magnitude measurement range is from 0.5 Ω to 30 Ω. The total power consumption of the integrated conditioner is 93.8 mW. The total input-referred noise of the circuit is shown in Figure 16. As can be observed, the system exhibits low noise at the frequencies of interest.

By analyzing the post-layout simulation results of the integrated conditioner, it can be concluded that its characteristics are significantly improved compared with the configuration in the discrete technology: the circuit nonlinearity, the dimensions, the power consumption, and the temperature sensitivity are much less than the prototype’s ones. In addition, the integrated conditioner can be used at different and higher frequencies since the cut-off frequencies of the differential amplifier and the precision AC-DC converter are much higher (than 8 MHz), while the BPF center frequency can be adjusted using the tuning circuit. In this research, the system operating frequency of 8 MHz was selected based on the optimal frequency of the measured sensors as well as on the requirements for the signal conditioner design. However, the proposed integrated conditioner can be used at higher frequencies and in other applications because of the tunable nature of the used filter.

Table 1 summarizes the performance of the proposed integrated signal conditioner and compares them with similar impedance measurement devices found in the literature. This system measures the smallest impedance values at the expense of higher power consumption. By looking at measurement time, it can be concluded that the proposed circuit is among the fastest conditioners, while the core area is among the smallest.

## 5. Conclusions

In this paper, a signal conditioner for variable impedance sensors is presented. It is designed using low-cost AMS 0.35 µm CMOS technology. The post-layout simulation results demonstrate that the circuit is suitable for measuring the sensor inductive impedance in the range of 10 nH to 550 nH (covering the measurement ranges of both tested displacement sensors). The conditioner measures the sensor impedance magnitude, which is sufficient for most variable impedance sensors since they are usually designed with one reactive component of the impedance. Although its primary (and here tested) purpose is the measurement of sensors with variable inductance (due to the measured inductive displacement sensor), the presented system can also be used for the measurement of sensors with variable resistive or capacitive output impedance. Because of its high dynamic response, it is possible to use this conditioner for the continuous measurement of fast response sensors.

The obtained results demonstrate that most of the characteristics of the integrated conditioner have been improved compared with those of the discrete realization. In addition, although the applied method represents a simplified version of the current–voltage vector methods and lock-in amplifiers, its application resulted in a system whose characteristics in terms of system dynamic response and temperature sensitivity are similar to the laboratory systems, though with smaller accuracy and measurement range. However, the designed system has much smaller dimensions, power consumption, and design complexity. Its limitation is the application only where it is sufficient to measure the modulus of sensor output impedance. Additionally, the proposed integrated signal conditioner measures the smallest impedance values and has a very fast response and small area compared with similar signal conditioners found in the literature.

The system performance could be further improved by integrating digital blocks, which can, in addition to miniaturization of the system, significantly increase its system possibilities in terms of expanding the measurement range and controlling the elements of the analog block, thus adjusting the measurement characteristics.

## Figures and Tables

**Figure 1 sensors-24-06501-f001:**
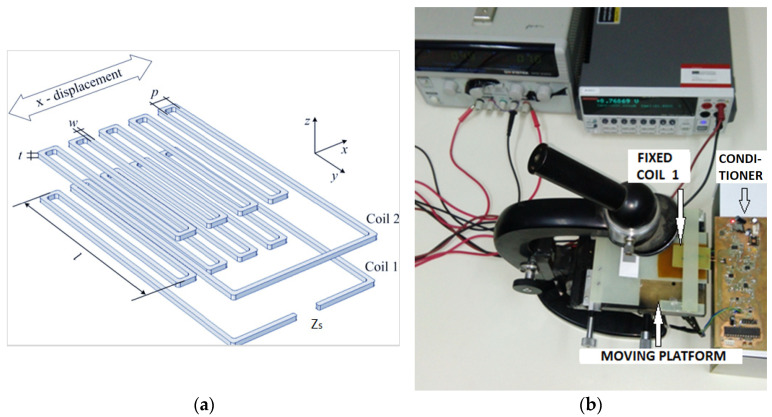
(**a**) Sensor element for the detection of x-displacement. (**b**) Measurement setup with the discrete signal conditioner.

**Figure 2 sensors-24-06501-f002:**
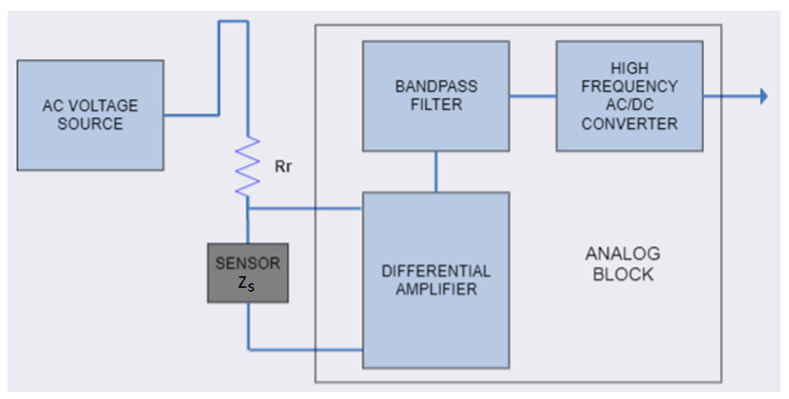
Conditioner block diagram.

**Figure 3 sensors-24-06501-f003:**
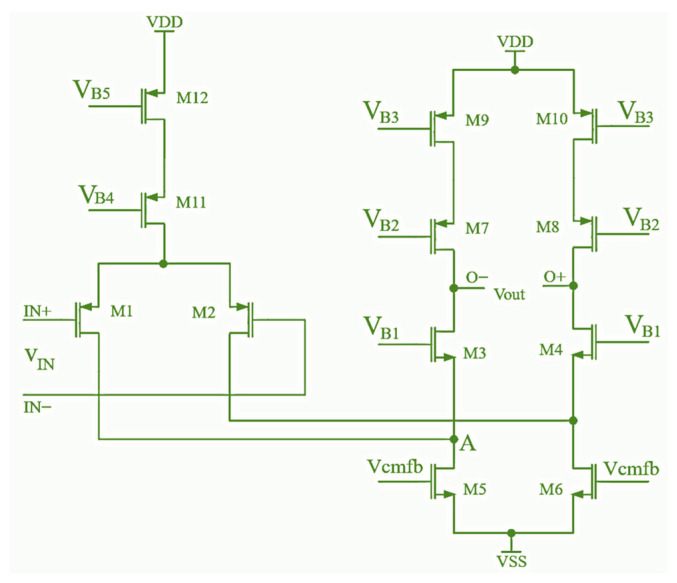
Schematic of the fully differential folded amplifier.

**Figure 4 sensors-24-06501-f004:**
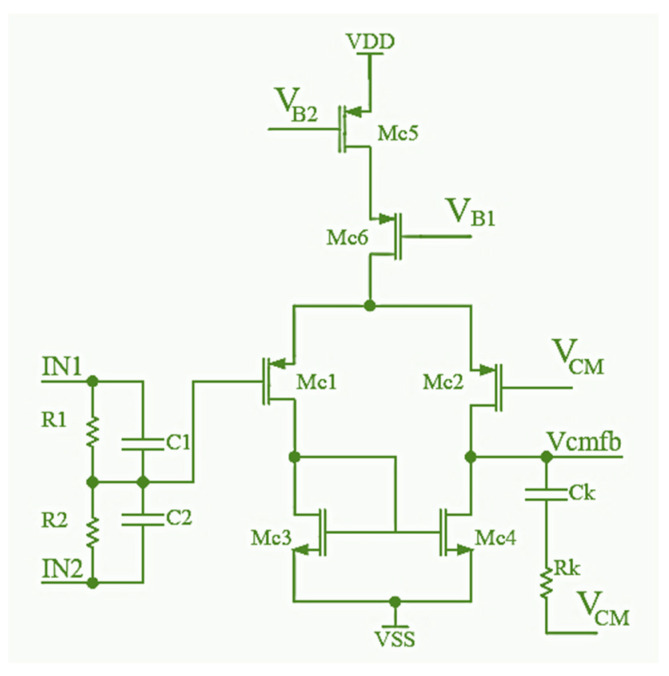
Schematic of the CMFB circuit.

**Figure 5 sensors-24-06501-f005:**
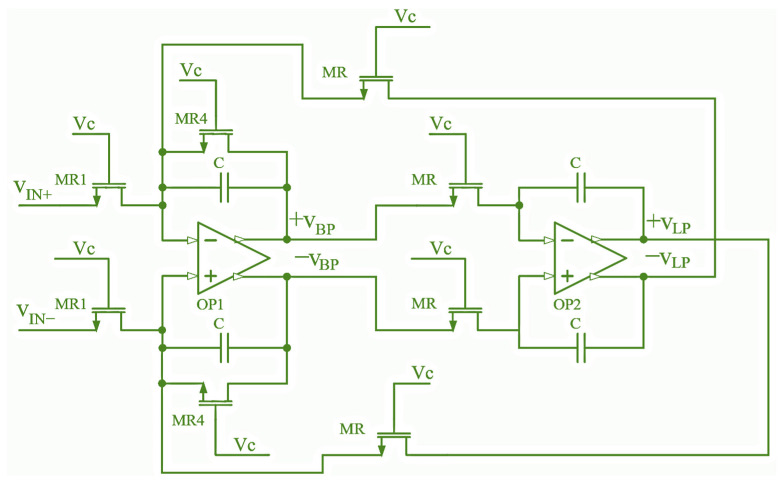
Schematic of the differential Tow–Thomas biquad filter in MOSFET-C topology.

**Figure 6 sensors-24-06501-f006:**
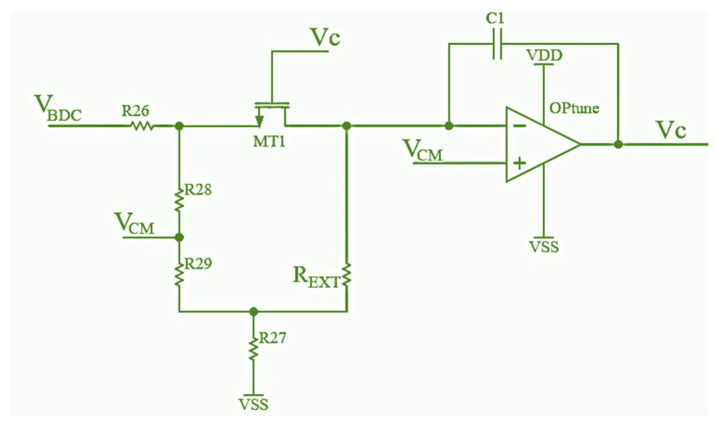
Schematic of the filter tuning circuit.

**Figure 7 sensors-24-06501-f007:**
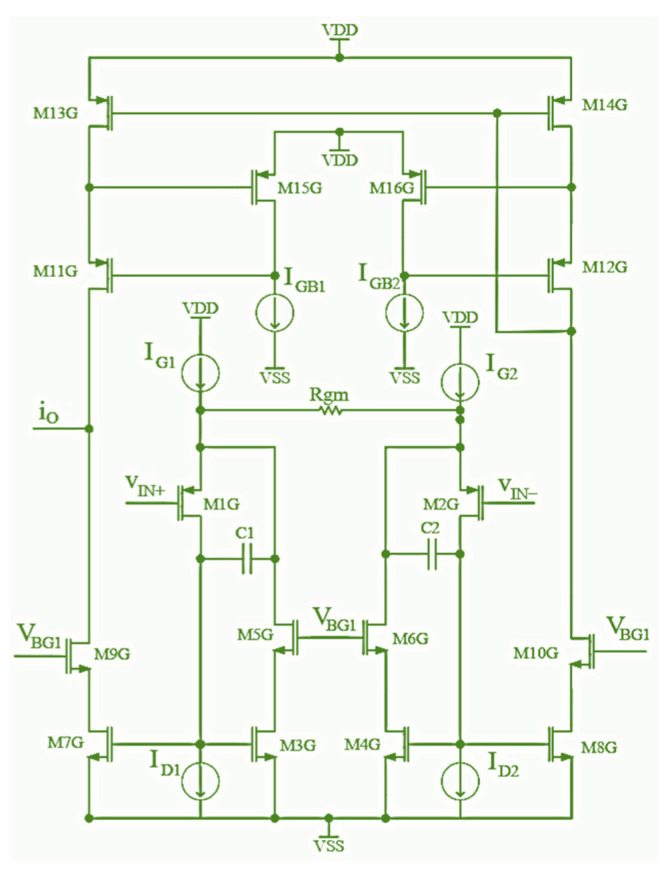
Schematic of the proposed Gm voltage current converter (conveyor).

**Figure 8 sensors-24-06501-f008:**
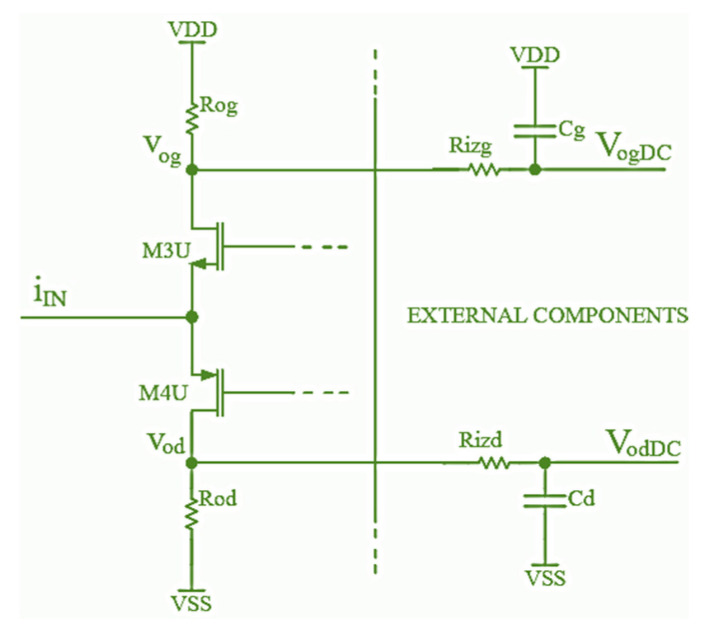
Schematic of the CMOS AB current rectifier.

**Figure 9 sensors-24-06501-f009:**
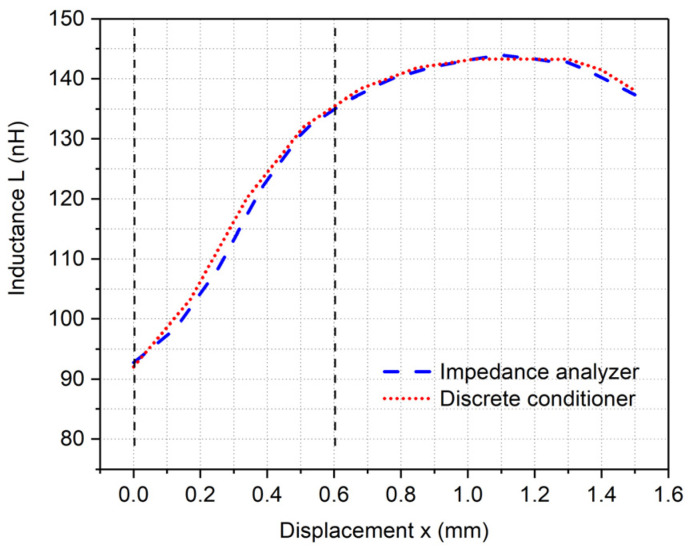
Comparison of the conditioner discrete realization measurement to the impedance analyzer measurement.

**Figure 10 sensors-24-06501-f010:**
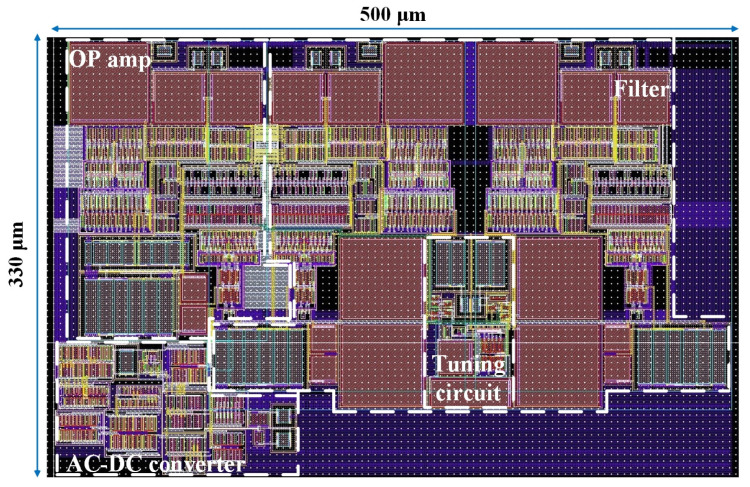
The integrated conditioner layout.

**Figure 11 sensors-24-06501-f011:**
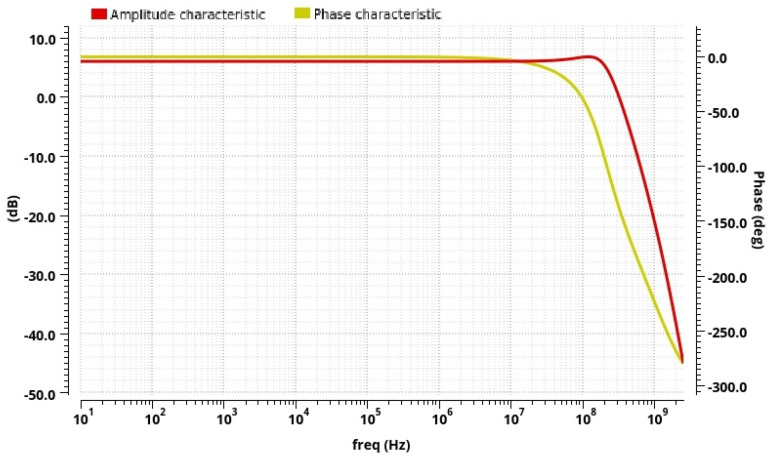
Post-layout simulation results: amplitude and phase characteristics of the designed operational amplifier in a closed-loop (gain of 2) configuration.

**Figure 12 sensors-24-06501-f012:**
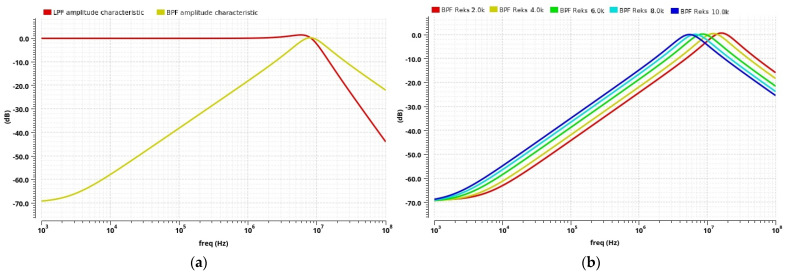
Post-layout simulation results: (**a**) LPF and BPF amplitude characteristics, (**b**) control of the BPF center frequency by the external resistor, and (**c**) BPF amplitude characteristics at the temperature change.

**Figure 13 sensors-24-06501-f013:**
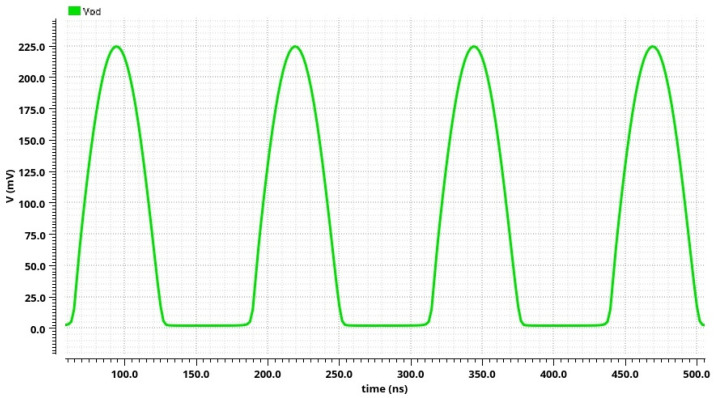
Post-layout simulation results: V_od_ output signal of the CMOS AC-DC converter.

**Figure 14 sensors-24-06501-f014:**
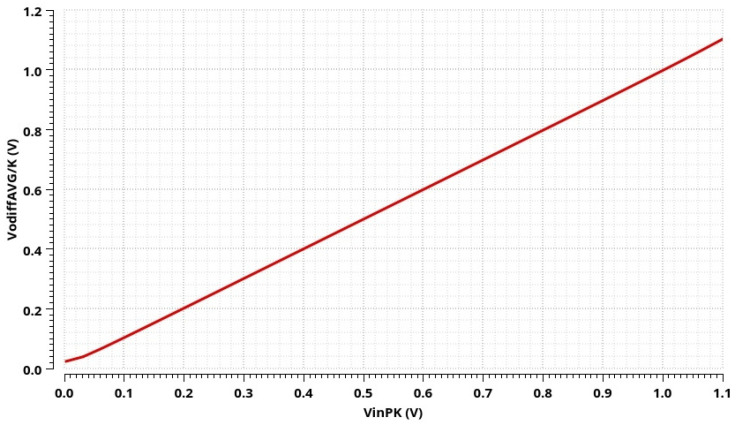
Post-layout simulation results: transfer function of the proposed AC-DC converter.

**Figure 15 sensors-24-06501-f015:**
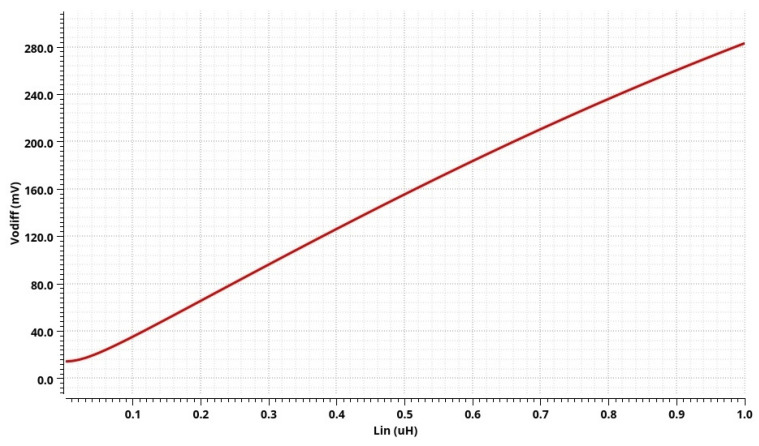
Post-layout simulation results: dependence of the integrated conditioner output voltage on the input impedance.

**Figure 16 sensors-24-06501-f016:**
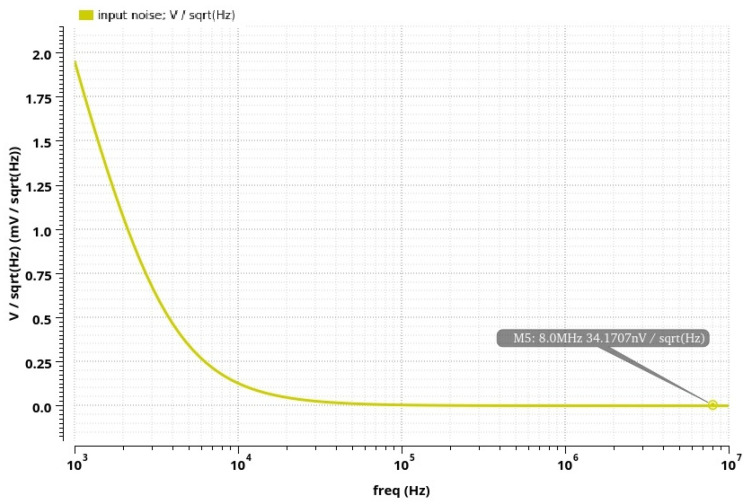
Post-layout simulation results: the integrated conditioner input-referred noise.

**Table 1 sensors-24-06501-t001:** Performance comparison of the signal conditioners.

Parameter	[2]	[18]	[21]	[15]	[23]	This Paper
Architecture	sine- fitting	analog lock-in amplifier	resonant with square wave mixers	LC resonant oscillator	sine lock-in	simplified current- voltage measurement
Operating frequency (kHz)	1	115	100 to 4000	12,000	up to 1000	8000
Measurement range	Z: 2 kΩ to 8 kΩ	R: 100 Ω to 10 kΩ	L: 100 nH to 3 μH	Z: 300 Ω to 1100 Ω	32 Ω to 5.3 kΩ	L: 10 nH to 550 nH Z: 0.5 to 30 Ω
Error (%)	0.85	1.90	9	3	0.04	1.2
Power consumption (mW)	N/A	0.885	10	N/A	3.4 per channel	93.8
Technology	discrete	0.18-µm CMOS	0.35-µm CMOS	discrete	0.35-µm CMOS	0.35-µm CMOS
Measurement time (ms)	500	N/A	<0.100	N/A	N/A	<1
Core area (mm^2^)	N/A	0.075	3.5	N/A	1.52	0.165
Supply voltage (V)	±12	1.8	5	N/A	±2.5	3.3

## Data Availability

The data supporting the findings of this study are available from the first author or corresponding author upon reasonable request.

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
