# Peer review of "Integrated Precision High-Frequency Signal Conditioner for Variable Impedance Sensors"

_sensors, 2024, doi:10.3390/s24206501_

Round 1

Reviewer 1 Report

Comments and Suggestions for Authors

1          This manuscript describes in detail the design of a system to measure linear displacement by using an inductive sensor. The impedance of the sensor is measured using a bipolar measurement at a single frequency.  The design of the signal conditioner is well described. Discrete and integrated designs are compared.

2          The introduction describes various methods of measuring impedance that have been used elsewhere. I was a little surprised that no reference is made to the very large literature on biomedical impedance measurements. Both bipolar and tetrapolar measurements have been used in Electrical Impedance Tomography and Electrical Inductance tomography. There are also many commercial systems that measure whole body and segmental impedance. Multi-frequency measurements are used to separate the impedance changes that result from intracellular and extracellular components of tissue.

3          The system designed seems appropriate to the particular displacement sensor described. However, the impedance measurement is made at a single frequency and provides impedance magnitude only..  

4          Possibly the title and certainly the Abstract should say that an inductive linear displacement transducer design is described.

5          There are a lot of figures. Some, for example Fig 7, could be removed

6          The high temperature sensitivity is a significant disadvantage. Why has this not been investigated and design changes made? 5.2% over 0 to 70C. Is this a linear change with temperature? What would the sensitivity change over a room ambient change of say 15-30C be?

7          Fig 10. What is the cause of the positive to negative sensitivity change around 1mm displacement?

8          It is a little disappointing that after a very detailed design assessment the final paragraph leaves the impression that many further  improvements are needed.      

Author Response

  1. Summary

Thank you very much for taking the time to review this manuscript.

Your comments are all valuable and have been very helpful in revising and improving our paper. Please find the detailed point-by-point responses below and the corresponding corrections highlighted/marked in red in the re-submitted manuscript.

We hope the revised manuscript will be regarded as an improvement in terms of quality, clarity, and relevance.

  1. Point-by-point response to Comments and Suggestions for Authors

Comments 1: This manuscript describes in detail the design of a system to measure linear displacement by using an inductive sensor. The impedance of the sensor is measured using a bipolar measurement at a single frequency. The design of the signal conditioner is well described. Discrete and integrated designs are compared.

Response 1: Thank you very much for the positive comments. Although the postlayout simulation results only at 8 MHz are presented in the manuscript, the integrated signal conditioner can be used at different frequencies. Details are provided within the response to your Comments 3. It is our mistake as we have not emphasized clearly this advantage in the manuscript. This is corrected in the revised manuscript.

Comments 2: The introduction describes various methods of measuring impedance that have been used elsewhere. I was a little surprised that no reference is made to the very large literature on biomedical impedance measurements. Both bipolar and tetrapolar measurements have been used in Electrical Impedance Tomography and Electrical Inductance tomography. There are also many commercial systems that measure whole body and segmental impedance. Multi-frequency measurements are used to separate the impedance changes that result from intracellular and extracellular components of tissue.

Response 2: Thank you very much for the professional feedback and suggestions. Yes, we agree with you. There are various types of impedance measurement sensors and they have a wide range of applications. As our research focuses on the displacement sensor, in the introduction, we provided an overview of similar measurement systems only. Impedance measurement ranges in biomedical applications usually differ from those used in this paper. According to your comment, we modified the section Introduction by adding new references and the following text (related to electrical impedance tomography):

In electrical impedance tomography (EIT) by measuring real and imaginary components of the body impedance, many aspects of a person's health status can be found. In EIT impedance conditioners, many variations of synchronous detectors are used, to eliminate various sources of noise. These conditioners usually measure impedance from around hundreds of Ω to hundreds of kΩ [22], which is the range of bioimpedance of human tissues. The measurement range is mostly in the range of kHz (where A/D converters are usually used instead of synchronous detectors to find impedance), but for cancer diagnosis it can go up to several MHz [23].

  1. Pennati, F.; Angelucci, A.; Morelli, L.; Bardini, S.; Barzanti, E.; Cavallini, F.; Conelli, A.; Di Federico, G.; Paganelli, C.; Aliverti, A. Electrical Impedance Tomography: From the Traditional Design to the Novel Frontier of Wearables. Sensors202323, 1182. https://doi.org/10.3390/s23031182.
  2. P. Kassanos, L. Constantinou, I. F. Triantis and A. Demosthenous, An Integrated Analog Readout for Multi-Frequency Bioimpedance Measurements, in IEEE Sensors Journal, 2014, vol. 14, no. 8, pp. 2792-2800.

In addition, we have added new references [15] and [18] to be used for performance comparison in Table 1.

Comments 3: The system designed seems appropriate to the particular displacement sensor described. However, the impedance measurement is made at a single frequency and provides impedance magnitude only...

Response 3: Thank you very much for pointing this out. It seems we did not emphasize clearly enough that the proposed integrated conditioner can be used at different frequencies. Your comment allows us to provide a clearer explanation. The main focus of this research was the design and testing of a measurement system for a particular displacement sensor. There was no need for measurement at several frequencies, and single-frequency impedance magnitude measurement was sufficient to obtain precise measurements in the displacement sensor we tested. Since the operating frequency of the conditioner discrete realization was 8 MHz (the skin effect has a significant influence in planar PCB traces and thus the displacement sensor at higher frequencies), the postlayout simulation results of the integrated conditioner were also given at 8 MHz (for comparison purpose). However, as explained in the paper, the bandwidth of the used amplifiers and AC-DC convert is much wider and the proposed filter is tunable, thus with very small modifications the proposed conditioner can be used at different frequencies, higher or lower than 8 MHz. By changing the value of the external resistor in the tuning circuit, the band-pass filter center frequency can be adjusted (please see Fig. 13(b) in the old version of the manuscript and Fig. 12(b) in the revised manuscript) and thus the system operating frequency can be changed. The following sentences in the manuscript state that the integrated signal conditioner can be used at different frequencies.

However, the proposed integrated conditioner can be used at higher frequencies and in other applications. Because of the tunable nature of the used filter, it can be easily adjusted to different operating frequencies.

We emphasized this advantage several times in the revised manuscript (marked in red).

In addition, one sentence in the Abstract was modified to clarify that the integrated signal conditioner can be used at different frequencies.

The operating frequency in this case was 8 MHz, but the circuit can be easily adjusted to different operating frequencies (due to the tunable filter).

Comments 4: Possibly the title and certainly the Abstract should say that an inductive linear displacement transducer design is described.

Response 4: Thank you very much for this suggestion. As explained above, the proposed integrated conditioner can be used at higher frequencies and in other applications. In this paper, it was tested to measure the inductive impedance of the displacement sensor. Therefore, we believe that there is no need to change the paper title. But we agree with you that we should add more clarification to the abstract. Therefore, the Abstract has been changed by adding the following sentences.

First, an inductive linear displacement sensor design is described and the signal conditioner discrete realization is presented. Based on this system's requirements, the integrated conditioner is proposed.

Comments 5: There are a lot of figures. Some, for example Fig 7, could be removed.

Response 5: Thank you for this suggestion. Yes, we agree with you. The Fig. 7 shows no specific detail. We modified the manuscript by deleting Fig. 7 and making the necessary changes.

Comments 6: The high temperature sensitivity is a significant disadvantage. Why has this not been investigated and design changes made? 5.2% over 0 to 70C. Is this a linear change with temperature? What would the sensitivity change over a room ambient change of say 15-30C be?

Response 6: Thank you for this insightful observation. As noted in the manuscript, high sensitivity to temperature variations is estimated as a consequence of a simple filter topology used in this prototype (the signal conditioner discrete realization). We investigated this problem and decided not to make further changes in discrete realization but to design an ASIC system with improved performance. This was one of the reasons for developing the integrated system, where we solved this problem with the filter tuning circuit. This circuit provides less temperature sensitivity (in addition to adjusting the filter center frequency).

Comments 7: Fig 10. What is the cause of the positive to negative sensitivity change around 1mm displacement?

Response 7: Thank you for this question. The response is related to the design or behavior of the sensor tested. The sensor's linear response is around 0.6 mm, and outside of this range, the sensor has a somewhat sinusoidal response, which is not useful for displacement measurement. We changed Fig 10 to emphasize this behavior, and added the following sentence in section 4.1 of the revised manuscript:

The linear response of the tested sensor for the displacement measurement is around 0.6 mm, as shown with a vertical dotted line in the figure.

Comments 8: It is a little disappointing that after a very detailed design assessment the final paragraph leaves the impression that many further improvements are needed.

Response 8: Thank you very much for the constructive comment and suggestion. It seems we left a wrong impression about the main goal of our research and the recommendations we gave in the section Conclusions. Our research started with the investigation of the displacement sensors so the specification for the conditioner discrete realization was based on its behavior. We kept the same specification for the integrated version but obtained improved performance (as expected) satisfying all the needed requirements. The integrated conditioner is suitable for measuring the sensor inductive impedance in the range from 10 nH to 550 nH, which covers the measuring range of both types of measured displacement sensors.

In the end, we have provided guidelines for further improvement of the device's characteristics so it can be used in other applications and for other measurement ranges.

To be more consistent, we have modified significantly the section Conclusions. The part you have referred to was changed to the following sentence.

The system performance could be further improved by integrating digital blocks which can, in addition to the miniaturization of the system, significantly increase the system possibilities in terms of expanding the measurement range and controlling the elements of the analog block and thus adjusting the system measurement characteristics.

We very much appreciate your positive marks for General Evaluation and Quality of English Language.

We have expanded the Section Introduction by adding new references as you recommended.

Reviewer 2 Report

Comments and Suggestions for Authors

The main issue with this paper is the absence of experimental results, as the findings are based entirely on simulations. The paper concentrates solely on inductance and does not adequately demonstrate how the ASIC enhances impedance measurement accuracy, as claimed by the title. Impedance systems usually involve a combination of resistance, inductance, and capacitance, so this narrow focus may not be sufficient. It might be more appropriate to consider publishing this paper with additional experimental data to support the claims.

Comments on the Quality of English Language

The writing style lacks conciseness and tends to be redundant.

Author Response

  1. Summary

Thank you very much for taking the time and effort to review and evaluate this manuscript.

Please find the detailed point-by-point responses below and the corresponding corrections highlighted/marked in red in the re-submitted manuscript.

We hope the revised manuscript will be regarded as an improvement in terms of quality, clarity, and relevance.

  1. Point-by-point response to Comments and Suggestions for Authors

Comments 1: The main issue with this paper is the absence of experimental results, as the findings are based entirely on simulations. The paper concentrates solely on inductance and does not adequately demonstrate how the ASIC enhances impedance measurement accuracy, as claimed by the title. Impedance systems usually involve a combination of resistance, inductance, and capacitance, so this narrow focus may not be sufficient. It might be more appropriate to consider publishing this paper with additional experimental data to support the claims.

Response 1: Thank you for the insightful observations that helped us to improve the manuscript quality. It seems that we have not comprehensively presented our work. We started the research by investigating the displacement sensors so the specification (and presented results) for both, the discrete and the integrated, conditioner realization was based on its behavior. The inductive linear displacement sensor design was described in detail and the experimental results of the signal conditioner discrete realization were presented. In addition, we have proposed the integrated signal conditioner obtaining improved performance (as expected) and satisfying all the tested system needed requirements. We agree with you that the quality of the manuscript would be much better if we could provide experimental results for the integrated conditioner as well. Unfortunately, we had no resources to fabricate the proposed design. However, based on our previous experience and the fact that we used an old/higher process node (not an advanced node with short-channel transistor behavior and dominant second-order effects) and lower operating frequencies, it can be expected that the experimental results would not differ significantly from the postlayout simulation results.

Moreover, we updated section 4.1 in the revised manuscript by adding the explanation that the measured displacement sensor also has nonzero resistance and explaining its influence on the Ls measurement.

The measurements were made with a smaller variant of the displacement sensor having the inductance Ls value in the range from 50nH to 100nH. This sensor impedance Zs also consists of the resistive part Rs, which changes from 0.62 Ω to 0.67 Ω with displacement in the linear measurement range. By taking the fixed mean value of Rs equal to 0.65 Ω, the maximal error from the average value of the resistance in calculating Ls from Zs is only 0.35%.

Although we focused on inductive measurement in this paper, the proposed conditioner can also measure sensors with variable output resistance or capacitance based on the proposed research.

Thank you very much for the mostly positive General Evaluation

We have expanded the Section Introduction by adding new references and descriptions to provide a more comprehensive overview.

Regarding your requirement for moderate editing of the English language and comment that the writing style lacks conciseness and tends to be redundant, we proofread thoroughly the manuscript, deleted and changed some sentences, re-wrote some phrases, and made some small changes to enhance the manuscript's quality. We agree that the Introduction and Conclusions sections should be presented in clearer and more concise language. So we've modified those sections the most. All changes regarding the English language are marked in red.

Reviewer 3 Report

Comments and Suggestions for Authors

The impedance of sensors is variable in many commercial devices and industrial systems. The manuscript presented an integrated signal conditioner which is consisted of an amplifier, a tunable band-pass filter and a high-frequency AC-DC converter. The topic is interesting but there are some issues that need to be considered as follows.

1.      The power consumption seems to be high. It is necessary to provide some discussions. For example, the contribution of each block.

2.      The conditioner discrete realization measurement results show good agreement with the impedance analyser measurement results. It is suggested to add simulation results in Figure 10.

3.      Only post-simulation is provided without tapeout. The floorplan of layout can be improved and each block should be identified in the figure.

4.      A table should be added to compare your work with prior related works. It is very important to demonstrate the performance of the proposed circuit.

Author Response

  1. Summary

Thank you very much for taking the time to review this manuscript.

Your comments are all valuable and have been very helpful for revising and improving our manuscript. Please find the detailed point-by-point responses below and the corresponding corrections highlighted/marked in red in the re-submitted manuscript.

We hope the revised manuscript will be regarded as an improvement in terms of quality, clarity, and relevance.

  1. Point-by-point response to Comments and Suggestions for Authors

The impedance of sensors is variable in many commercial devices and industrial systems. The manuscript presented an integrated signal conditioner which is consisted of an amplifier, a tunable band-pass filter and a high-frequency AC-DC converter. The topic is interesting but there are some issues that need to be considered as follows.

Comments 1: The power consumption seems to be high. It is necessary to provide some discussions. For example, the contribution of each block.

Response 1: Thank you very much for requesting a clearer explanation of the circuit power consumption. Yes, we agree with you that the power consumption seems to be high. Unfortunately, it is mainly the result of a trade-off between the circuit's power consumption and bandwidth (operating frequencies) in the used technology. A low-cost old/higher process node offers a higher voltage swing but increases power consumption due to higher power supply voltage. Furthermore, as it provides lower ft and fmax values (limitations), we had to increase power to enable higher operating frequencies. The power consumption of each block was already given in the submitted manuscript. However, those values were obtained for the separately tested blocks. Due to shared polarization in the system, the values provided in the revised manuscript are a bit lower.

The total power consumption of the operational amplifier is 21.54 mW, since the DC current values had to be in the order of mA to obtain high open loop gain at 8 MHz and higher frequencies (i.e. high GBWP).

The filter power consumption is 42.88 mW, as it contains two operational amplifiers. While Gm-C filters are a very popular choice in CMOS IC design because of their low consumption and high working frequency, they have a high nonlinearity, which is the main reason why they were not used in this circuit.

The power consumption of the precision AC/DC converter is 29.38 mW. The DC currents in this circuit also had to be in the order of mA to obtain significant bandwidth.

We updated the revised manuscript by adding the above-provided clarification for the high power consumption in Section 4.2. All changes are marked in red.

Comments 2: The conditioner discrete realization measurement results show good agreement with the impedance analyser measurement results. It is suggested to add simulation results in Figure 10.

Response 2: Thank you for this recommendation. Yes, we agree with you that it would be interesting to compare the conditioner discrete and integrated realization results with the impedance analyzer measurement results on one graph. The data provided in Figure 10 is acquired by measuring the physical sensor, and unfortunately, we couldn’t make similar measurements with the integrated signal conditioner since we only have postlayout simulation results for the inductance measurement and can’t obtain dependence on the displacement (x-axis in Figure 10).

Comments 3: Only post-simulation is provided without tapeout. The floorplan of layout can be improved and each block should be identified in the figure.

Response 3: Thank you for pointing this out. Yes, we agree with you that the research impact would have been increased if we had fabricated the IC and provided experimental results. Unfortunately, we had no resources to fabricate the proposed design. However, based on our previous experience and the fact that we used an old/higher process node (not an advanced node with short-channel transistor behavior and dominant second-order effects) and lower operating frequencies, it can be expected that the experimental results would not differ significantly from the postlayout simulation results.

Regarding your comment about the layout floorplan, we agree with you that it can always be better. But, in this case, it seems to have been a misunderstanding. Due to the low resolution (quality) of the IC layout photo, it appears that there is a lot of empty space in the layout where transistors are actually located. All differential transistors have been matched using the common-centroid technique and resistors have been matched using the interdigitated technique. We kept blocks/elements as close as possible to have as small an area as possible, but at the same time, we had to provide a good symmetry. The good agreement between the postlayout and the schematic-level simulation results proved the good layout design.

In the revised manuscript, we provided the higher-resolution layout photo (still some ”fake” empty space can be seen) with the block identification according to your recommendation. In addition, we have extended the power network at the bottom and on the right side of the IC. Please see Fig. 10 in the revised manuscript.

Comments 4: A table should be added to compare your work with prior related works. It is very important to demonstrate the performance of the proposed circuit.

Response 4: Thank you very much for this suggestion. The revised manuscript was updated with Table 1 presenting the signal conditioner performance comparison. We also added some new references ([15], [18], [23]) in the section Introduction that have been used for the FoM comparison.

We updated the revised manuscript by adding the following text related to the results comparison.

Table I summarizes the performance of the proposed integrated signal conditioner and compares them with similar impedance measurement devices found in the literature. This system measures the smallest impedance values, at the expense of higher power consumption. By looking at measurement time, it can be concluded that the proposed circuit is among the fastest conditioners, while the core area is among the smallest.

Table 1 Performance comparison of the signal conditioners.

Parameter

[2]

[18]

[21]

[15]

[23]

This work

Architecture

sine-

fitting

analog lock-in amplifier

resonant with square wave mixers

LC resonant oscillator

sine lock-in

voltage

measurement

Operating

frequency (kHz)

1

115

100 to 4000

12000

up to 1000

8000

Measurement range

Z: 2 kΩ to 8 kΩ

R: 100 Ω to 10 kΩ

L: 100 nH to 3 µH

Z: 300 Ω to 1100 Ω

32 Ω to 5.3 kΩ

L: 10 nH to 550 nH

Z: 0.5 to 30 Ω

Error (%)

0.85

1.90

9

3

0.04

1.2

Power

 consumption (mW)

N/A

0.885

10

N/A

3.4 per

channel

93.8

Technology

discrete

0.18-µm CMOS

0.35-µm CMOS

discrete

0.35-µm CMOS

0.35-µm CMOS

Measurement time (ms)

500

N/A

<0.100

N/A

N/A

<1

Core area (mm2)

N/A

0.075

3.5

N/A

1.52

0.165

Supply

voltage (V)

±12

1.8

5

N/A

±2.5

3.3

We very much appreciate your positive marks for General Evaluation and Quality of English Language.

Round 2

Reviewer 2 Report

Comments and Suggestions for Authors

This paper is lack of testing results, pls add testing results before the submission

Comments on the Quality of English Language

N/A

Author Response

Point-by-point response to Comments and Suggestions for Authors

Comments 1: This paper is lack of testing results, pls add testing results before the submission.

Response 1: Please accept our deep apologies if we were not clear enough in our previous response. The paper presents only the experimental results of the signal conditioner discrete realization. Unfortunately, we don't have the funds to manufacture (fabricate) the proposed integrated signal conditioner. Since the IC is not fabricated, we are not able to provide the experimental results.

However, from our experiences in testing the IC we have designed and fabricated so far, it can be expected that the experimental results would not differ significantly from the postlayout simulation results presented in the paper.

Considering your feedback about moderate editing of the English language required, we have engaged a Full Professor of English at the Faculty of Technical Sciences to do the proofreading. All changes are marked in red in the revised manuscript.